# Multi-Actor Partnerships for Agricultural Interactive Innovation: Findings from 17 Case Studies in Europe

Susana B. Guerrero-Ocampo [1], José M. Díaz-Puente [1,*] and Juan Felipe Nuñez Espinoza [2]

1  Escuela de Ingeniería Agroalimentaria y Biosistemas (ETSIAAB), Universidad Politécnica de Madrid, Avda. Puerta de Hierro 2, 28040 Madrid, Spain
2  Escuela Superior de Ingenierías Industrial, Aeroespacial y Audiovisual de Terrassa (ESEIAAT), Universidad Politécnica de Cataluña, Carrer de Jordi Girona, 31, 08034 Barcelona, Spain
*  Correspondence: jm.diazpuente@upm.es; Tel.: +34-672-279-218

**Abstract:** Innovation is widely regarded as a key factor for the economic development and competitiveness of companies and countries. It is, therefore, widely considered a policy instrument in various sectors, such as agriculture. In this sector, agricultural innovation is seen as a systemic and interactive phenomenon, which is the result of interactions between innovators and knowledge-generating organisations, as well as social and economic aspects of the context. This paper studies the social structures of multi-actor partnerships involved in interactive innovation processes in agricultural innovation systems, analysing the type of actors involved and the roles they play in the innovation process. For this purpose, 17 case studies were analysed in the framework of the Liaison project, an H2020 project, using social network analysis (SNA) and descriptive statistics. The results show that the studied multi-actor partnerships have been mostly funded by outside sources of funding, highlighting European funds. The innovation networks have a heterogeneous composition, but when we analyse the frequency of interactions there is a tendency to establish greater interaction between organisations that are of the same type. In the "core" of innovation networks, research entities and farmers are central actors with the main role of technician expert and case study field workers, respectively.

**Keywords:** agricultural; multi-actor partnerships; SNA; interactive innovation; case studies

## 1. Introduction

Innovation is widely regarded as a key factor for the economic development and competitiveness of companies, regions, and nations alike [1,2]. Furthermore, it is considered the heart of value creation and a key strategy to improve productivity for rural development [3,4]. For these, the European Commission has emphasised the incorporation of innovation in its various policy instruments. The EC considers innovation to be "the renewal and enlargement of the range of products and services and the associated markets; the establishment of new methods of production, supply and distribution; the introduction of changes in management, work organisation, and the working conditions and the skills of the workforce" [5] (p. 23). This concept, however, has acquired nuances over time, depending on the experience and lessons learned during the implementation of the promoted policy instruments.

In the European Union (EU), the "Agricultural Knowledge and Innovation System" (AKIS) model is defined as 'a concept that seeks to encompass and influence the complexity of knowledge and innovation processes in the rural sphere' [6] (p. 7). The AKIS covers aspects such as formal institutional linkages between public and private institutions and/or informal knowledge networks among farmers, how information and knowledge flows (and how innovation takes place), and how these processes can be strengthened [6].

According to Fieldsend et al. [7], the concept of AKIS was mainly operationalised into policy by the Strategic Working Group (SWG) on AKIS of the Standing Committee

of Agricultural Research (SCAR), in consultation with the EC's Directorate-General for Agriculture and Rural Development (DG AGRI). Consequently, the European Innovation Partnership for Agricultural Productivity and Sustainability (EIP-AGRI) was introduced as a tool to accelerate innovation in agriculture, forestry and rural development and to create synergies between different policy programmes at both the EU and the Member State level with relevant results.

EIP-AGRI promotes the "interactive innovation model", which is defined as the collaboration among several actors to co-create knowledge between practice, scientists, advisers, enterprises, NGOs, etc., taking into account different dimensions (including technical, organisational, and social aspects), which helps to bridge the gap between science and practice, applying a "systems approach". This means that farmers, farm advisors, scientists and other actors collaborate throughout the project to develop innovative solutions to practical problems. These solutions have a greater chance to be relevant and used, as they are developed with and for farmers or practitioners [8]. According to Ingram et al. [9], in order to achieve interactive innovation it is necessary to create a social space in which learning and knowledge sharing are combined through innovation networks that bring together different actors, with different visions and forms of knowledge.

It is relevant to point out that the diffusion of innovations refers to a non-deterministic process, which depends on a diversity of endogenous and exogenous factors that continuously change, which makes it possible to be present in different areas of knowledge and technology. The diffusion of innovations has been analysed by various actors, with Everett Rogers being one of the most relevant through his theory of "Diffusion of Innovations" [10]. However, this theory has controversial points that have been identified by other actors, such as [11] and Albrecht (1963), who question, among other aspects, the categorisation of adopters established from 'innovators' through to 'laggards'. Without sufficiently taking into account the complexity of the social, cultural (Singh, 2003), historical, and structural contexts of populations, especially in rural environments, the decision to adopt and implement an innovation is made individually but is executed as a social, communal and multidimensional action, determined by the presence of actors, social structure and system of linkages [12]. Additionally, the adoption of innovations, coupled with their diffusion, is a process of internalisation and reproduction of a certain idea in the individual and in the social system, which implies a long maturation time in the human group [10,13]. Therefore, the diffusion and adoption of innovations can serve as a conceptual tool for accessing complex social realities and for setting up complex systems in the management of capacities within these realities [14].

In the agricultural sector, innovation is increasingly seen as a systemic and interactive phenomenon, which is the result of social and economic development, as well as interactions between innovators and knowledge-generating organisations [15]. Additionally, innovations encompass technological and non-technological changes, resulting from the exchange and recombination of knowledge between various actors. This knowledge is acquired through "learning by doing" (based on innovation capacities) or through learning "from others" (based on social network) [16].

In the agricultural innovation systems approach, a network is defined as an innovation space where actors (individuals and organisations) interact with one another and are connected in some way. Social networks can define, limit, or enhance an individual's opportunities for social learning by influencing membership or participation in a given innovation process, thereby affecting access to knowledge [17]. The structure of social networks and the characteristics play a crucial role in the circulation of knowledge [18] so that a better understanding of the structures and functioning of the actor network would provide the basis for modifying existing knowledge-sharing mechanisms in agricultural innovation systems to improve performance.

In this direction, considering the agricultural innovation as a systemic and interactive phenomenon is relevant analysis of social structures of multi-actor partnerships involved in interactive innovation processes in agricultural systems. For this, we analysed 17 case

studies of interactive innovation initiatives related to agriculture in European rural areas, identified under the umbrella of a H2020 project that aimed to optimise interactive innovation within the scope of rural projects and their networks.

The purpose of this paper is, therefore, to examine, through a series of case studies, the networks that promote agricultural interactive innovations in European rural areas. To achieve this, we use social network analysis (SNA) and descriptive statistics to respond the following questions: (i) In which European areas are the most resources that enable interactive agricultural innovation concentrated? (ii) What is the composition and ties of networks according to the organisation leading the multi-stakeholder partnerships for agricultural innovation? (iii) What types of organisations make up these multi-stakeholder partnerships and what roles do they play? Ultimately, we aimed to propose novel lines of action that could prove useful to consolidate the networks of actors and the process of agricultural innovation in rural areas.

## 2. Analytical Framework

### 2.1. Multi-Actor Partnerships for Innovation and Leadership

Multi-actor partnerships are cross-sector partnerships with partners from three sectors (business, public, and civil society), created to address a priority issue for multiple partners [19]. Multi-actor partnerships potentially create an opportunity for partners to access more resources, such as knowledge, financial support, and social capital, overcoming the limitations of a single organisation or sector (Kuenkel and Aitken, 2015) but have diverse challenges, such as managing the diverse interests of multiple partners and keeping them active. Kochan (2016) remarks that partnerships are prone to crisis and disappointment, and a key factor in the resiliencies the ability of multiple partners to apply the 'tools of the partnership' to address the challenges as they arise. Among those tools are the skills and leadership of key actors [19].

According to Lambrecht et al. [20], to decrease the uncertainty inherent in an innovation process, numerous contacts are seen as particularly helpful, especially via a certain person who facilitate the links with the different actors. This will increase the chance of discovering crucial opportunities. This concept is often referred to as 'innovation broker', whose main purpose is to build appropriate linkages in innovation systems and facilitate multi-actor interaction in innovation.

### 2.2. Innovation Networks: Composition and Ties

An innovation network is considered to be a set of connections between people with diverse social relationships in which information, knowledge, and other social processes flow, facilitating the innovation process [21]. Networks composed of partners with heterogeneous experiences will be in a better position to benefit from the present experiences than networks composed of partners with homogeneous experiences, and they will, therefore, make better decisions [20]. Networks with a high degree of homophily may limit access to information and knowledge dissemination outside the closed circle of the network, as links to actors outside the network are limited. Network actors with high homophily often have links to people who are similar to themselves (e.g., farmer-to-farmer interaction) [18] and allow for the consolidation of ideas and concepts.

Additionally, Bogers (2011) suggests that any innovation network has two layers. The first layer, or 'core group', is small, with actors who work in close collaboration and communication, sharing knowledge openly. The second one consists of a 'larger periphery' of diverse actors that are less involved, though they participate in the innovation process. With these actors, not all information is shared, but it helps them to gain legitimacy and support capacity for the innovation [22].

Another aspect to be taken into account in networks is the quality of ties between actors. There are two different positions on this issue. On the one hand, the "strength of weak ties" theory introduced by Granovetter (1973) states that weak ties (connections involving low investments in terms of time, affection, intimacy, and reciprocity) are more

likely than "strong" ties to connect distant nodes in a network, thus, transmitting new information to actors [23]. Thus, unique and non-redundant information is more readily available through an individual's informal acquaintances than through close friends [17,24]. On the other hand, there is the so-called channel bandwidth thesis, proposed by Aral and Van Alstyne (2011), which postulates that strong ties (based on higher levels of solidarity, trust, and emotional involvement, with people interacting more frequently) are, in fact, likely to transmit more consolidated information than weak ties, both in terms of quantity and quality [23]. Both types of ties could be seen as contributing to the innovation process, with weak ties contributing to the diffusion of innovations (new information), while strong ties contribute to the adoption of innovations (same information).

### 2.3. Actor's Role, Innovation and Knowledge

In several pieces of work, the role of local actors has been analysed to explain the relationship between innovation processes and the diffusion of knowledge as the innovative "medium", innovation systems at national and regional level (Cooke, 1998; Lundvall et al., 2002; Malerba, 2002, 2010), and the Triple Helix (Leydesdorff, 2000, 2005; Viale, and Pozzali, 2010). Two key elements stand out in these models that are closely linked: networking and multilevel governance (control of the processes of knowledge generation and diffusion), both of which significantly influence the evolution of innovation systems. In this sense, the capacity of the networks of actors to build a local system permeable to innovations and to develop, at different scales, an interactive process of promotion, creation, and management of knowledge is crucial for the good performance of local systems and innovations [25].

Some studies identified different forms of knowledge and innovation. Explicit knowledge refers to codified knowledge, which can be systematised, written, stored, and transferred, whereas tacit knowledge is described as implicit, local, context dependent, inherently intangible, and results from talents, experience, and abilities, created through individual experience. Both forms of knowledge are complementary, and one knowledge form may transform into another form through different types of interaction [26]. Regarding the modes of innovation, there are two types: (i) the science, technology, and innovation (STI) mode that is based on a formal process for generating and using explicit knowledge; and (ii) the doing, using, and interacting (DUI) mode in which tacit knowledge or know-how is acquired through an informal learning process. Combining both modes leads to improved innovation capabilities. However, there is still a bias to consider innovation processes largely as the STI mode [27].

In addition, knowledge can be acquired formally (e.g., between consultants and professional and scientific bodies) or informally (e.g., experiential knowledge acquired through everyday interaction) [28]. Experiential learning is a constant process that happens not only at the individual level but also at the interpersonal level, as practical experiences are shared and joint problem solving is undertaken, in accordance with social learning concepts [18].

### 2.4. Social Network Analysis (SNA)

There are several methods to analyse and evaluate agricultural innovation from a systems approach, either from a static or dynamic point of view [29]. Among them, social network analysis (SNA) stands out as a useful method from an infrastructural and static perspective of innovation support systems, mapping institutional linkages, visualising relationships between actors, and assessing the prominent position of actors within the system (in terms of centrality, number of linkages, strength of linkages) [17].

SNA is a tool for analysing and representing the structure of social networks, through matrices and network diagrams, as well as mathematical measures [30], based on the principles of graph theory, in order to determine the presence, direction and strength of connections between actors in a network. SNA allows measuring an individual's access to communal resources, based on the position he or she occupies in the network, which

partly determines the constraints and opportunities he or she encounters [31]. SNA argues that the relationships or connections between people in a network are fundamental conduits through which many types of resources pass: knowledge, information, advice, materials, etc. [32].

In the context of innovation, SNA provides an understanding of how actors interact, how information and resources move between and among them, and how agent's roles and relationships are structured. Data for SNA are commonly based on measurements of relationships between actors and sets of actors, in addition to the attributes of individual actors [17].

## 3. Materials and Methods

Four methodological steps were followed in this study. The first was the selection of agricultural innovation case studies, previously analysed in the framework of a European project focused on interactive innovation optimisation to accelerate innovation in agriculture, forestry, and rural development. Subsequently, we proceeded to identify each of the participating actors, taking into account the interactions between them, actor types and roles they played in each innovation initiative. The third step was the evaluation of the identified interactions and the analysis of the data using the SNA, which was complemented with descriptive statistics. Finally, the data analysed were visualised to facilitate the process of presentation and discussion of the findings.

### 3.1. Case Study Selection

Our data and findings were obtained in the context of a European research project that aimed to make a significant contribution to optimising interactive innovation project approaches and the implementation of European Union (EU) policies to accelerate innovation in agriculture, forestry, and rural development. The LIAISON project (Better Rural Innovation: Linking Actors, Instruments and Policies through Networks) was funded by the Horizon 2020 (H2020). Through this project we identified an extensive database of interactive innovation projects and programs across Europe, from which 200 initiatives were selected for desk research [33]. Subsequently, a sample of 30 case studies (CS) was selected and analysed along 2020, with 283 in-depth interviews conducted with members of 30 multi-actor partnerships, using a common analytical framework [33].

From these 30 in-depth case study reports, we select the cases for the analysis, considering the following criteria: (i) the innovation initiatives operate in the agricultural sector; (ii) the core of the alliance was made up of more than one type of organisation; and (iii) availability of information on the actors that participated in the multi-actor partnerships. This resulted in a final selection of 17 case studies (Table A1).

### 3.2. Data Collection in the Case Studies

In each case study, we proceeded to identify each of the participating actors, taking into account the relationships between them, actor types and roles they played in each innovation initiative. This process was carried out, taking into account two steps (i) identification of actors by interactions and (ii) identification of actor types and roles, which are described in detail as follows.

Step I: Identification of actors by interactions in each case study. We used the analytical framework in-depth case studies proposed by Cronin et al. [33] to analyse the relationships that take place between the actors linked in the interactive innovation process. This framework considers five types of interactions: with funding mechanisms, along interactions within the partnership, external actors and the context/environment, and societal challenges. We identified the actors considered in each interaction in the 17 case studies selected. The last type of interaction is not considered in this study because it is not related directly with actors and their interactions but with the impact of the innovation initiatives on societal challenges.

The interactions used in the study are described below:

- Interaction 1: Funding Mechanisms. This interaction is related to funding mechanisms of the innovation initiative. The actors identified in this interaction are related to the grant awarding process (e.g., actors involved in writing proposal), the provision of funding (e.g., funder, co-funder, etc.) and its management.
- Interaction 2: Core of multi-actor partnership. This interaction is related to the core of multi-actor partnership and the creation of the innovation initiative. The actors included in this interaction are considered the "core" of the initiative and have actively participated in the co-creation of the innovation initiative, assuming diverse and complementary roles.
- Interaction 3: Networking with external actors. This interaction relates to the network of actors external to the core of the innovation initiative. The actors included in this interaction know the innovation initiative and have contributed in a specific way from their expertise. They have not participated in the whole process of "co-creation" of the innovation initiative; however, their contributions have contributed to its development.
- Interaction 4: Interaction with the context. This interaction is related to the context in which the innovation initiative takes place. The actors identified have been influenced or influence the innovation initiative but are not aware of it and have not been directly involved in it.

Step II: Identification of actor types and roles. Using the analytical framework in-depth case studies, we identified the type of actors in their roles in each case study. It is important to indicate that actors are considered to be individuals or institutions/organisations linked in the interactive innovation process. Within institutions or organisations there may be different departments or individuals, representing the same institution in which case the institution is considered as an actor but with different roles, depending on its representatives. Ten types of actors are identified in the agricultural innovate on networks:

- Administrative bodies: any governmental agency or organisation charged with managing and implementing regulations, laws, and government policies (e.g., local municipality, regional government, national government, ministries, departments, EU institutions, etc.)
- Civil society: citizens who individually or collectively carry out activities independently of governmental structures, political parties, businesses, and religious institutions (e.g., NGOs, local community groups, LEADER groups, etc.)
- Educational institutions: institution primarily engaged in educating others, through the process of teaching—learning and disseminating knowledge, e.g., primary education, (agricultural) schools, universities in their role as educator, etc.
- Farmers: a farmer is a person engaged in agriculture, who raise living organisms (plants or animals) for food or raw materials (e.g., pioneer farmer, organic farmer, etc.).
- Market actors—demand side: persons, institutions, or organisations—that demand the goods and services related to the innovation initiatives, e.g., business, processing or marketing SME, processing or marketing producer organisation, retailers, consumers and their organisations, other companies, etc.
- Market actors—supply side: persons, institutions, or organisations—that offer goods and services related to the promoted innovation initiatives, e.g., business, suppliers, manufacturers, service providers, etc.
- Research entities: institutes or universities primarily engaged in research related to agricultural issues, in their role of research institution.
- R&D departments in companies: areas of companies dedicated to R & D & I activities for the development of innovations in the agricultural field with a market approach.
- Support organisations: persons or institutions that provide the necessary resources for the effective and efficient operation of innovation initiatives, e.g., management advisors, financial actors (banks, venture capital, business angels), network organisations, etc.
- Others: any type of actor that does not fit with the types from above.

Actor's role is referring to the function that an actor plays in the interactive innovation process. Fourteen different roles are identified in the agricultural innovation networks, although not all roles are present in each of the innovation networks. An actor can play more than one role in the interactive innovation process.

- Administrative manager: responsible for operational and support activities. Maintains communication with the different partners.
- Advisor: provides specific advice to the innovation initiative (networking, accounting issues, market issues, certifications, among others).
- Case study field workers: provides practical knowledge in the innovation process. Applies innovation proposals in the field and provides feedback on them, based on their application and experience.
- Civil servant: manages and implements government regulations, laws, policies, and programmes that affect the context in which the proposed innovation takes place.
- Co-funder: it provides smaller financial resources for the implementation of the innovation initiative. It can also provide valued resources (e.g., infrastructure, human resources, etc.).
- Communication and dissemination: disseminate the results of the innovation initiative.
- Competitor: offer on the market a good and/or service, similar to the one developed by the innovation initiative.
- Coordinator: leads the innovation initiative. Coordinates the work and communication between the different partners of the innovation initiative.
- End-user: individuals or institutions/organisations who ultimately use or are intended to use a product or service.
- External supervisor: it externally monitors the implementation of policies and instruments for the proper management of the innovation initiative.
- Funder: it provides increased financial resources for the implementation of the innovation initiative.
- Funding management body: manages financial resources on behalf of the funding organisation ensuring compliance with relevant regulations.
- Technician expert: provides theoretical knowledge in the innovation process. Participate during the implementation of innovation proposals in the field in order to check their validity and make the necessary modifications.
- Supplier: provides goods and/or services necessary for the implementation of innovation initiatives.

### 3.3. Data Analysis

SNA was applied to construct the social structure networks of 17 case studies. To this extent, a bimodal matrix was created using the actors' ratings by interactions and roles. The value established by each interaction among actors is considering the number of times the type of actor has been identified in each interaction, taking into account the role played. All interactions identified received a score of 1.

The elements of SNA used in this study are synthesised in (Table 1). At network level, we analysed the network size. At node level, we analysed the centrality measure of degree. The centrality measures were obtained using the software UCINET 6 version 6.735.

In addition, descriptive statistics were used to complement the analysis of the composition and ties of the multi-actor partnerships.

**Table 1.** Elements of social network analysis used in the study.

| Element | Definition |
| --- | --- |
| Node | Any individual, organisation, or other entity of interest. |
| Tie | Links between nodes, which denote interactions. |

**Table 1.** *Cont.*

| Element | Definition |
| --- | --- |
| Network | Graphical representation of relationships that displays points as nodes and lines ties; also referred to as a graph. |
| Centrality | Structural attribute of nodes in a network determined by their position in the network; centrality measures include degree, closeness, and betweenness. |
| Degree | Number of ties a node has to other nodes (strategic access to net information). |
| Indegree | A number of ties going into a node (social legitimacy). |
| Core | Cohesive subgroup within a network in which the nodes are connected to the maximum. |
| Periphery | Nodes that are only loosely connected to the core and have minimal or no ties among themselves. |

Source: [34–37].

*3.4. Data Visualisation by SNA*

The explanatory power of the network visualisation is quite good for explaining the structural positions of actors [38]. Graphs provide an effective approximation of the network structure and reinforce the information obtained through the analysis of the centrality measures. Nodes and ties have visual properties that can be mapped to provide with key network information. The size of the node shows the degree centrality of the actor, and the thickness of a line is used to indicate the strength of a tie. In addition, type of organisation and roles are represented in the graph using different forms and colours.

The graphs were obtained using software Visone 2.19 and UCINET 6 version 6.735 (NetDraw 2.178). The graph designed with Visone has a distribution on the "y" axis of the graph, so that the more central the node is, the location is on the top, and the graphs elaborated with UCINET have a distribution on the "x" axis of the graph, so that the more central the node is, the location is to the right.

**4. Results and Discussion**

The results are presented around the main characteristics of agricultural innovation networks analysed: (i) geographical concentration of resources, (ii) composition and ties of networks, according to the leader of multi-actor partnerships and (iii) types of organisations and roles of multi-actor partnerships for agricultural innovation.

*4.1. Geographical Concentration of Resources*

The 17 case studies vary in dimension, with some having a local (24%), sub-national (47%), and multinational dimension (29%) (Table A1). Taking into account the geographical concentration of resources, it was found that countries with the greatest capacity to obtain resources for the development of agricultural innovation initiatives are Germany, France, United Kingdom, Spain, and Belgium, followed by Italy and Greece according to in-degree centrality (Figure 1). This geographical distribution of resources could be linked to the European rural development policy implemented through rural development programmes (RDPs) drawn up by EU Member States and regions, which set out priority approaches and actions to address the needs of the specific geographical area they cover. During the period 2014–2020, 118 national and regional RDPs were implemented, co-financed by the European Agricultural Fund for Rural Development (EAFRD) and national contributions (https://enrd.ec.europa.eu/policy-in-action/rural-development-policy-figures/rdp-summaries_en, accessed on 30 August 2022). It is worth noting that in this study the main funder is Common Agricultural Policy/Rural Development Program (CAP/RDP) (Table A1), and the five countries identified for their high ability to capture more resources for agricultural innovation have both RDPs at national (5) and regional level (63 in total—Germany 13, France 27, United Kingdom 4, Spain 17 and Belgium 2) while other countries mostly only have RDPs at national level. Additionally, ability to capture more resources for agricultural innovation in these areas, could be related to the high capacity of the "core" of multi-actor partnerships to formulate innovation proposals to funding agencies, previous

experience of partners, the alignment of the innovation proposals with the priorities, and compliance with the requirements of the funding call, among others.

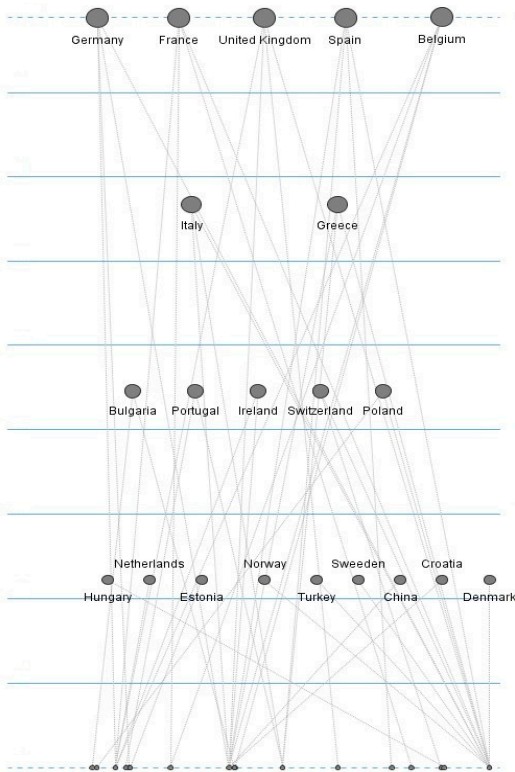

**Figure 1.** Geographical concentration of resources, according to indegree centrality (Visone).

*4.2. Composition and Ties*

It was found that each multi-actor partnership has different composition in terms of actor number involved the type of organisation´s participants and roles that actors assumed in the agricultural innovation initiative. The 17 multi-actor partnerships studied consisted of 442 actors in total. The number of actors involved is varied, with multi-actor partnerships being most frequent, comprising a range of 32 to 39 actors and a range of 8 to 15 actors (29% in each range), followed by a range of 16 to 23 actors. The muti-actor partnerships studied were mostly composed of 7 or 6 types of organisations (24% in each one), followed by 10 and 8 types of organisations (18% and 18%, respectively) (Table 2). We identify a diversity of actors, indicating a potential expandability and complexity of the sources driving interactive innovation in each multi-actor partnership. This favours the amount of knowledge sharing, which is positively correlated with the number of collaborative links an organisation has within the innovation network [39]. In agricultural initiatives, some studies have shown that the heterogeneous network of actors provides smallholders with a greater diversity of options for accessing information, inputs, credit or other resources, and how certain actors play key bridging roles in making these options available to smallholders, which potentially translates into a greater number of livelihood options and opportunities for smallholders [17].

In each multi-actor partnership, we identified a "core" group of actors that brings together the central actors of the network, leads the interactive innovation initiative, and promotes the adoption of innovations, while there are also other actors that are considered peripheral and play a key role in the diffusion of innovations. This structure has previously been noted as a characteristic of innovation networks [22], and it is considered that ongoing communication between the two layers can further optimise the innovation process [40]. The 'core' of multi-actor partnerships was mainly composed by four types of organisations (29%), followed by six, three, and two types of organisations (18%, 18%, 18%, respectively) (Table 2).

**Table 2.** Number of actors and type of organisations of 17 multi-actor partnerships.

| Items | Nr. | % | Case Studies |
|---|---|---|---|
| Actors | 40–47 | 12% | CS15, CS16 |
| | 32–39 | 29% | CS01, CS05, CS07, CS10, CS13 |
| | 24–31 | 12% | CS11, CS17 |
| | 16–23 | 18% | CS04, CS09, CS12 |
| | 8–15 | 29% | CS02, CS03, CS06, CS08, CS14 |
| Type of organisations of multiactor partnership | 11 | 6% | CS15 |
| | 10 | 18% | CS01, CS10, CS13 |
| | 9 | 6% | CS05 |
| | 8 | 18% | CS08, CS11, CS16 |
| | 7 | 24% | CS06, CS07, CS12, CS17 |
| | 6 | 24% | CS02, CS03, CS04, CS09 |
| | 4 | 6% | CS14 |
| Type of organisations of "core" multiactor partnership | 6 | 18% | CS07, CS11, CS14 |
| | 5 | 6% | CS06 |
| | 4 | 29% | CS01, CS03, CS09, CS10, CS17 |
| | 3 | 18% | CS04, CS08, CS16 |
| | 2 | 18% | CS02, CS12, CS15 |
| | 1 | 12% | CS05, CS13 |

Source: Own elaboration.

In the 17 multi-actor partnerships studied, the project leaders are often research institutions (47%) (CS06, CS07, CS08, CS11, CS13, CS14, CS16, CS17), followed by farmers (29%) (CS01, CS02, CS05, CS10, CS15). R&D departments in companies (CS12), supply side market actors (CS03), civil society (CS04), and the support organisation (CS09) each led a single multi-actor partnership (Table 3).

### 4.2.1. Partnerships Led by Research Entities

By jointly analysing the eight multi-actor partnerships of agricultural innovation led by research entities (47%), we found that the network structure was made up of 10 different types of organisations. However, the cases analysed show that research entities interact mostly with organisations of the same type (homophily) (24.5%), based on links and previous experience of joint work between universities and research centres on agricultural innovation issues. The interaction of universities with farmers is the second most frequent (19.3%) and responds to the development of applied research in the field to validate or test innovations. The third in frequency are administrative bodies (14.5%) represented by governmental agency or organisation charged with managing and implementing regulations, laws, and government policies (e.g., EU institutions that funder research projects). In the eight multi-actor partnerships led by research entities, we found less frequent interaction of research entities with market actors from supply side (13.6%), support organisations (11.2%), educational institutions (6.9%), others (4.4%), actors of civil society (4.2%), market actors from demand side (3.6%), and there is not interaction with R&D departments in companies. These findings indicate a large separation between research activities and the business sector, which could have a negative impact on the commercialisation of agricultural innovations that may emerge led by research entities. The disconnection between firms and academia had been identified in other studies, and it can be partially explained by the different structure of incentives. Whereas academia rewards peer-review articles, presentation at conferences, etc., firms are governed by problem solving incentives (Klenk and Wyatt, 2015 cited by [41].

### 4.2.2. Partnerships Led by Farmers

By jointly analysing the five multi-actor partnerships of agricultural innovation led by farmers (29%), we found that network structure is composed by 10 different types of organisations, but farmers interact mostly with support organisations that provide resources for

the effective and efficient operation of innovation initiatives, e.g., management advisors, financial actors (banks, venture capital, business angels), and network organisations, (19.5%), followed by farmers (18.6%) and administrative bodies (17.7%), highlighting government agricultural extension services, ministry of agriculture, local governments, and funders of innovation initiative.

**Table 3.** Composition and frequency of interaction by type of multi-actor partnership leader.

| Leader/Interactions by Type of Organisation | Administrative Bodies (%) | Civil Society (%) | Educational Institutions (%) | Farmers (%) | Market Actors— Demand Side (%) | Market Actors— Supply Side (%) | Others (%) | R&D Departments in Companies | Research Entities (%) | Support Organisations (%) | Total % |
|---|---|---|---|---|---|---|---|---|---|---|---|
| Research entities | 14.5 | 4.2 | 6.9 | 19.3 | 2.1 | 13.6 | 3.6 | 0.0 | 24.5 | 11.2 | 100 |
| CS06 | 27.3 | 0.0 | 4.5 | 13.6 | 0.0 | 18.2 | 0.0 | 0.0 | 13.6 | 22.7 | 100 |
| CS07 | 1.3 | 0.0 | 0.0 | 48.1 | 2.5 | 26.6 | 2.5 | 0.0 | 8.9 | 10.1 | 100 |
| CS08 | 24.1 | 3.4 | 20.7 | 3.4 | 0.0 | 6.9 | 3.4 | 0.0 | 24.1 | 13.8 | 100 |
| CS11 | 26.3 | 21.1 | 10.5 | 13.2 | 0.0 | 5.3 | 2.6 | 0.0 | 15.8 | 5.3 | 100 |
| CS13 | 14.0 | 6.0 | 6.0 | 10.0 | 8.0 | 4.0 | 14.0 | 0.0 | 34.0 | 4.0 | 100 |
| CS14 | 21.2 | 0.0 | 0.0 | 18.2 | 0.0 | 0.0 | 0.0 | 0.0 | 48.5 | 24.2 | 100 |
| CS16 | 8.5 | 4.3 | 4.3 | 2.1 | 2.1 | 25.5 | 0.0 | 0.0 | 40.4 | 12.1 | 100 |
| CS17 | 18.2 | 0.0 | 21.2 | 15.2 | 0.0 | 6.1 | 3.0 | 0.0 | 18.2 | 18.2 | 100 |
| Farmers | 17.7 | 9.7 | 4.0 | 18.6 | 5.8 | 8.4 | 5.3 | 2.2 | 8.8 | 19.5 | 100 |
| CS01 | 24.0 | 8.0 | 2.0 | 14.0 | 8.0 | 14.0 | 6.0 | 0.0 | 2.0 | 22.0 | 100 |
| CS02 | 5.9 | 0.0 | 0.0 | 35.3 | 11.8 | 5.9 | 0.0 | 23.5 | 0.0 | 17.6 | 100 |
| CS05 | 14.9 | 4.3 | 6.4 | 31.9 | 0.0 | 4.3 | 6.4 | 0.0 | 6.4 | 25.5 | 100 |
| CS10 | 22.0 | 8.0 | 6.0 | 12.0 | 2.0 | 10.0 | 4.0 | 0.0 | 16.0 | 20.0 | 100 |
| CS15 | 14.5 | 19.4 | 3.2 | 12.9 | 9.7 | 6.5 | 6.5 | 1.6 | 12.9 | 12.9 | 100 |
| Market actor—supply side CS03 | 9.5 | 9.5 | 0.0 | 28.6 | 0.0 | 28.6 | 0.0 | 0.0 | 14.3 | 9.5 | 100 |
| Civil society CS04 | 36.0 | 24.0 | 0.0 | 16.0 | 0.0 | 4.0 | 8.0 | 0.0 | 12.0 | 0.0 | 100 |
| Support organisations CS09 | 17.5 | 2.5 | 0.0 | 22.5 | 0.0 | 0.0 | 0.0 | 5.0 | 37.5 | 15.0 | 100 |
| R&D departments in companies CS12 | 12.5 | 12.5 | 0.0 | 34.4 | 0.0 | 6.3 | 0.0 | 28.1 | 6.3 | 0.0 | 100 |

Source: Own elaboration.

Additionally, in multi-actor partnerships of agricultural innovation led by farmers was found interaction with civil society (9.7%), research entities (8.8%), and market actors from the supply side (8.4%). Lower frequency of interaction is found in others (5.3%), educational institutions (4.0%), market actors from demand side (5.8%), and R&D departments in companies (2.2%).

These findings are similar to those found by [17] that show the central role of extension and related public services (such as agriculture and rural development offices and their development agents, local governments, government-backed credit, and savings institutions), and farmers' cooperatives in the innovation processes of rural smallholders. However, the potential contributions of other innovation systems actors—private industry, entrepreneurs, civil society, and so on—remain largely untapped, which is visible in this study as a low frequency of interaction with this type of actor. The frequent interaction with other farmers in these farmer-led multi-actor partnerships supports the innovation adoption process, as the importance of peer-to-peer relationships and experimentation at the farm level for the farmers' learning process has been repeatedly shown (Ingram 2010). Thus, the individual learning activity on the farm is accompanied and enhanced by a process of social learning and adoption of innovations [18].

### 4.2.3. Partnerships Led by Others

Four multi-actor partnerships were led by market actor from supply side, civil society, support organisations, and R&D departments in companies. We found that the network structure of each case study is made up of six different types of organisations. It should be noted that the findings on the case studies analysed do not reflect the current trend found on the emergence of the private agricultural input supply sector as a supplier and disseminator

of new technologies [42]. This could be due to the fact that these findings are not exclusively focused on the rural environment and are not developed as interactive innovations.

Given that the limited number of case studies is not considered sufficient to characterise the composition of networks led by these types of actors, we focus on only the in-depth detailing of the composition of two of them, one led by civil society and the other by R&D departments in companies:

The multi-actor partnership led by civil society, represented by NGO, (CS04) has the most interactions with administrative bodies (36%), followed by other actors of civil society (24%) and farmers (16%). In lower frequency of interaction are research entities (12%), others (8%), and market actors from supply side (4%). There is no interaction with R&D departments in companies. These findings are similar to those found by [17] who note that NGOs in innovation initiatives are closely linked to government entities and community-based organisations established under the auspices of NGO activities. In addition, NGOs are often linked not only to local public sector service providers, but also to a range of other actors beyond the immediate locality, such as research institutes and universities. Another study found that the dominance of NGOs and the lack of entrepreneurial capacity in innovation networks can hinder social learning and the development of innovations that are commercial and responsive to end-user needs [39], so it is desirable to foster greater articulation between different innovation sectors.

The multi-actor partnerships led by R&D departments in companies (CS12) has a strong interaction with farmers (34.4%), with whom they worked together to test and validate the innovation ideas proposed, followed by interaction with R&D departments in companies (28.1%) because the innovation proposed was constantly monitored by the company itself and involved its internal teams. The interaction with administrative bodies (12.5%) was through various public bodies in the agricultural sector and specifically related to water resources management and with civil society (12.5%) through the irrigation communities. Interactions found in lower frequencies include research entities (6.3%), which elaborated a specialised study about irrigation systems, and market actors from the supply side (6.3%). Specific R&D projects are often motivated by the practical problems posed by new products, processes, and user needs. In this case, the R&D departments combined their technical knowledge with practical knowledge of farmers conducting experiments and interpreting the results to attend the need of improving the irrigation of crop fields [27]. The successful outcome of this business initiative could be related to the use of mixed strategies that combine innovation strategies in both ITS mode (involving universities and their institutional R&D area) and DUI mode (involving farmers and civil society), bringing together both formal and informal knowledge.

### 4.3. Types of Organisations and Roles

In the 17 multi-actor partnerships studied, the actors developed diverse roles (8 on average). Some actors participate in more than one type of interaction with different roles, identifying 675 actor interactions in total.

#### 4.3.1. Analysis by Type of Organisation and Roles at Global Level

Using the SNA, the degree of centrality of the actors that make up the 17 multi-actor partnerships is analysed, taking into account the types of organisations and the roles they played. It was found that the central role played by the actors was the communication and dissemination role. The research entities and administrative bodies are central actors that frequently interact in innovation networks, playing differentiated roles. In the case of research entities, the role of advisor and technician expert stands out, and in the case of administrative bodies, the roles of civil servant and funder stand out. Subsequently, we find the presence of farmers with the roles of case study workers and end-users; the market actors from supply side with their role, such as supplier, and actors considered like "others" with a principal role, such as communication and dissemination. The support organisations have three main roles: communication and dissemination, advisor, and technician expert.

The actor civil society stands out for its role as end-user and its communication and dissemination. The actors, R&D departments in companies, educational institutions, and market actors from demand side, are considered peripheral actors (Figure 2).

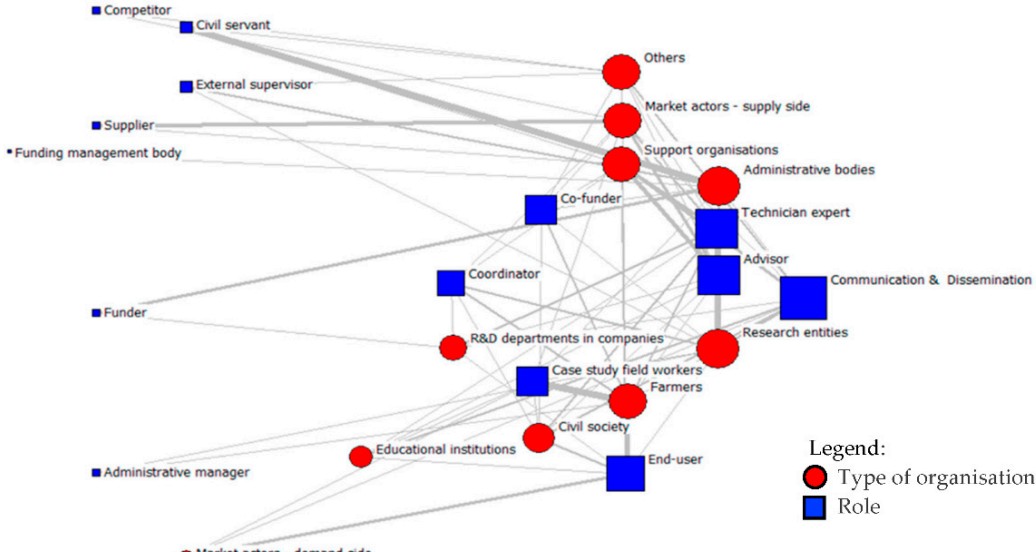

**Figure 2.** Type of organisation and roles in 17 multi-actor partnerships, according to degree centrality (UCINET).

4.3.2. Analysis by Type of Organisation and Roles According to Type of Interactions

In the 17 multi-actor partnerships, when we analyse each type of interaction, we found that there are variations in the degree centrality of actors.

Related to funding mechanisms (interaction 1), the central actors are administrative bodies with the role of funder mainly, followed by research entities with their roles of technician expert and coordinator, from which they contributed significantly to the formulation of the innovation initiative. Another important role in this type of interaction is co-funder, often played by farmers (Figure 3).

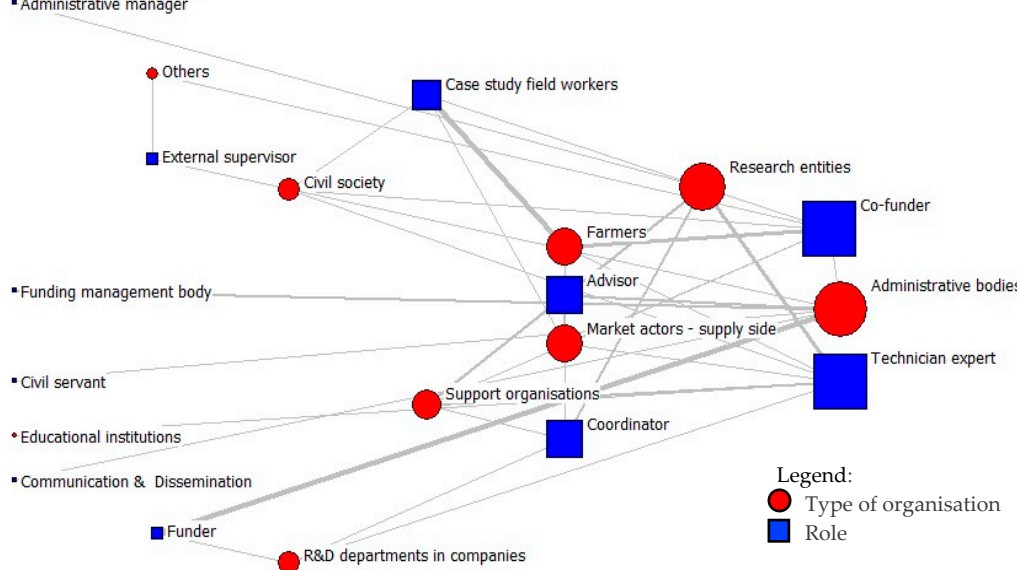

**Figure 3.** Type of organisation and roles by funding mechanisms, according to degree centrality (UCINET).

The 17 innovation initiatives have been partially or completely funded by outside sources of funding. Among administrative bodies identified, the main sources of funding for agricultural innovation initiatives are European funds. These include the Common Agricultural Policy/Rural Development Program (CAP/RDP) (29%), Horizon 2020 (12%), Interreg (12%), European Region Action Scheme for the Mobility of University Students (Erasmus +) (6%), and Life Program (6%). There are also initiatives whose main source of funding comes from miscellaneous funds (18%), and other innovation initiatives were funded by national funding, public (12%), and private (6%) (Figure 4). These findings are similar to those found by Esparcia [25] who identified that EU programmes (such as INTERREG, LEADER and SAPARD) were the most prominent sources of public funding for European innovation projects, although national and regional governments also provided substantial support, either directly or indirectly. In smaller and more modest projects, it also found a strong public presence in funding, both in the start-up and development phase of innovative projects.

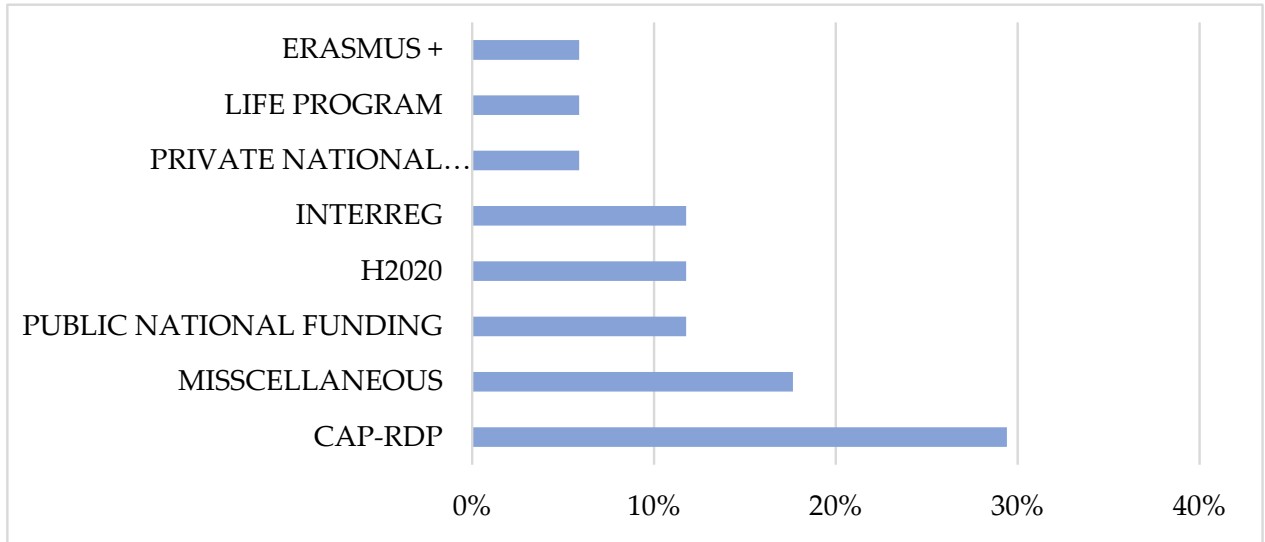

**Figure 4.** Main sources of funding for the 17 agricultural innovation initiatives.

In order to obtain funding for innovation initiatives, it is crucial to have actors that provide 'non-economic' support (it can be productive, corporate, institutional, social environmental, or a combination of these) [25]. In the 17 multi-actor partnerships, we found that research entities, with their roles of technician expert and coordinator, stood out for their significant contribution to the formulation of the innovation projects, due to their knowledge and experience, but other actors also contributed to a lesser extent. In relation to this, [25] did not find a clear pattern of "non-financial" support for innovation initiatives, as some projects combine a variety of these forms of support, each of which will be useful for different stages of the project but noted scientific support for the initial phases. The initial idea is often provided by the owners and/or managers (in our study considered "coordinators"), while in other cases, public bodies and NGOs provide key notions (explicit knowledge).

Related to the core of the multi-actor partnership (interaction 2), the central actors are farmers and research entities with the mainly role of case study field workers and technician expert, respectively. Another important role is advisor, often played by research entities, support organisations, and market actor from the supply side (Figure 5).

We found that at the core of the multi-actor partnerships there are both innovation and learning modes: STI mode (research entities, support organisations and market actor from supply side mainly, with the role of technical expert intervening throughout the innovation process or as an advisor providing specific and specialized advice in the innovation process) and DUI mode (farmers, testing, and validating the innovation proposal in the field), but

the network shows that the roles related to STI mode have greater centrality. In addition, it is considered necessary to deepen the analysis of both learning modes, which coexist and can complement each other, but this does not imply that they always work in harmony with each other. It is an important task for knowledge management to make the strong versions of the two modes work together to promote knowledge creation and innovation [18]. It is also important to note that several studies on agricultural innovation systems consider that a more inclusive and participatory process can prevent the common failure of innovations in the field. By including farmers in the innovation design process, their suggestions, needs, and knowledge are integrated, which favours the adoption and diffusion of new agricultural technologies [43].

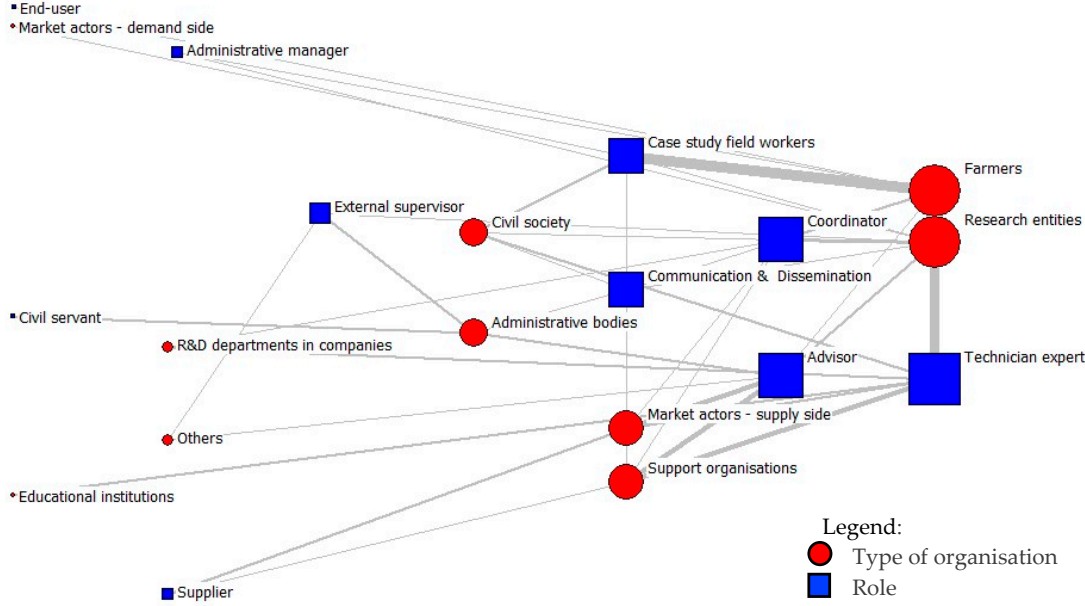

**Figure 5.** Type of organisation and roles by core of multiactor partnerships, according to degree centrality (UCINET).

Another important role in this type of interaction is coordinator, often played by research entities and farmers. This key role allows expanded social networks, increases trust, improves information flow among groups, sparks collaborative opportunities, helps to establish or maintain relationships that increase trust, and proactively overcomes innovation barriers [44].

Related to the relation with external actors of the multi-actor partnership (interaction 3), the central actors are research entities, followed by support organisations, administrative bodies, and others, that played the main role of communication and dissemination, advisors, and end-users of agricultural innovations (Figure 6). The central actors identified have contributed in a specific way to innovation, from their expertise, helping in its design, using it to validate or disseminating it. They have not participated in the whole process of "co-creation" of the innovation; however, they contributed to its development.

Related to the relation with the context (interaction 4), the central actors are support organisations and others, and the main roles are end-users and communication and dissemination (Figure 7). In this interaction, actors have been identified that are considered in the previously established typology but are not necessarily the same actors mentioned in the previous interactions, and their relationship with the innovation is referred to the use they make of the innovations or the dissemination of the innovations.

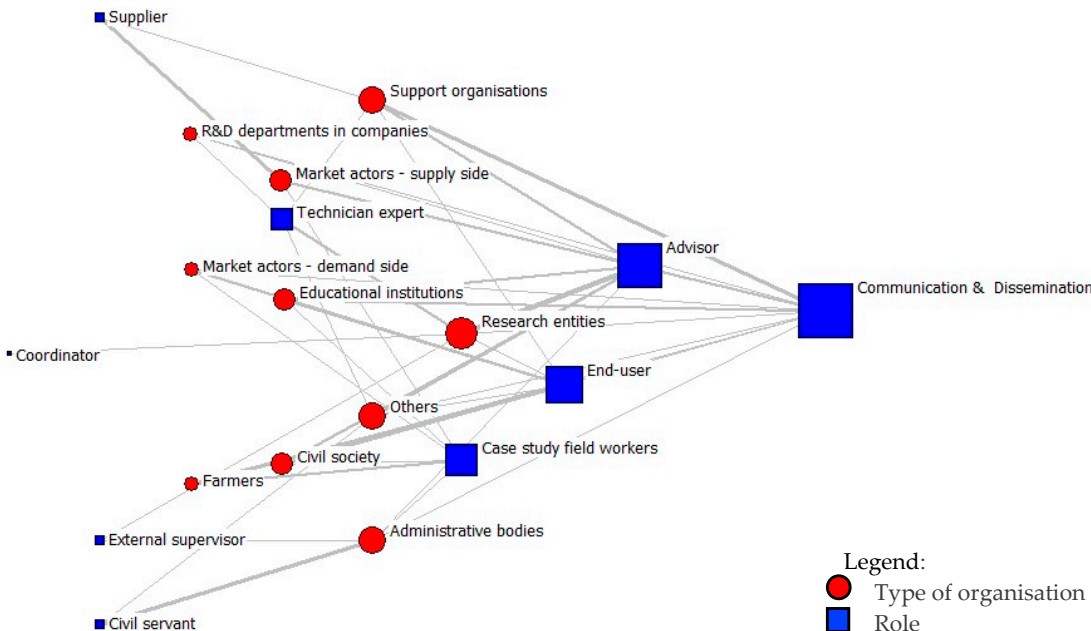

**Figure 6.** Type of organisation and roles by networking with external actors, according to degree centrality (UCINET).

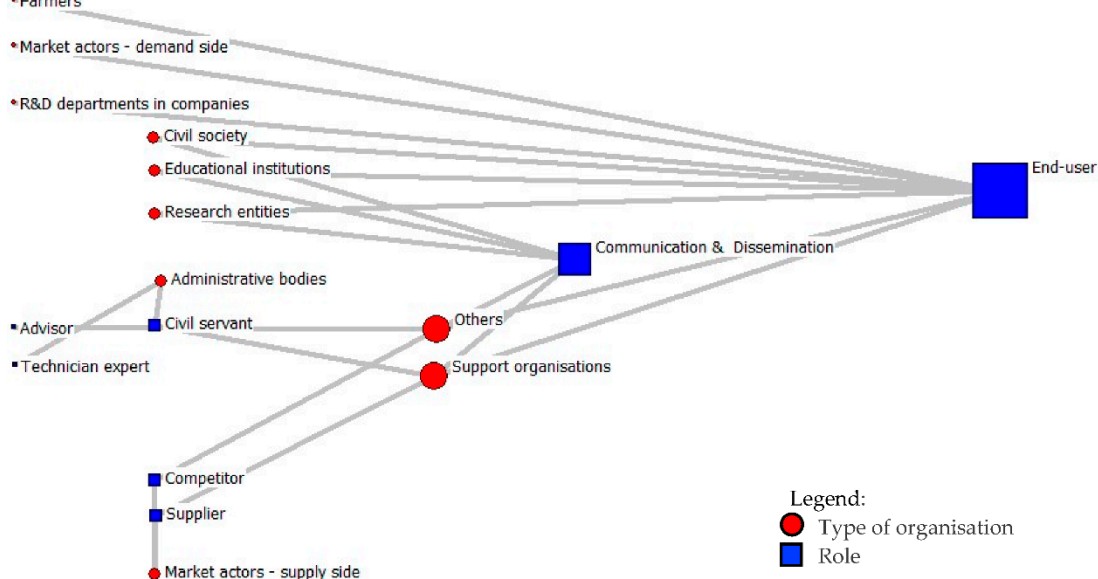

**Figure 7.** Type of organisation and roles by relation with the context, according to degree centrality (UCINET).

## 5. Conclusions

The main conclusions of this study are presented below.

Financial actors are a key element in the development and implementation of innovative projects in rural areas due to the dependence of many projects on public support (during and even after the final stages of innovation development).

The countries with the highest concentration of resources for agricultural innovation have planning instruments aligned with rural development policy, both at the national and subnational levels, allowing them to establish priority approaches and actions to address the specific needs of the various geographical areas in their countries. This decentralised planning not only contributes to increasing the opportunities for attracting more financial resources, but also allows the development of local capacities for the formulation and

implementation of innovation initiatives in rural areas, reducing the gaps between rural and urban areas, and could contribute to reducing the differences between the most and least developed countries in Europe.

The innovation networks analysed have a heterogeneous composition, but an analysis of the frequency of interactions shows a tendency towards more interaction between organisations of the same type (homophily). In the multi-actor partnerships led by universities and the market actor on the supply side, interaction with actors of the same typology is the most frequent. In the multi-actor partnerships led by farmers, civil society, and R&D departments of companies, interaction with actors of the same typology is the second most frequent.

We identified a diversity of actors and their roles in the innovation process, according to the four interactions analysed. In relation to the funding mechanisms, the central actors are mainly the administrative bodies with the role of funders, followed by the research entities with their roles of technical expert and coordinator who contributed significantly to the formulation of the innovation initiative, and the farmers as co-funder of the innovation initiatives. At the core of the multi-stakeholder partnership, the central actors who participated in the "co-creation" of innovations are farmers and research entities with the main role of case study field workers and technical experts, respectively. Regarding to the relation with the external actors of the multi-stakeholder partnership, the central actors are research entities, followed by support organisations, administrative bodies, and others, who have contributed in specific ways to the innovation, from their expertise, helping in its design, using it to validate it or disseminate it. Finally, related to the relation with the context, the central actors are the support organisations and others, and their relationship with the innovation is referred to the use they make of the innovations or their diffusion.

The case studies analysed have a greater tendency of STI modes of innovation and learning but successful innovation initiatives show the use of mixed strategies that combine innovation strategies in both STI and DUI modes of innovation and learning, bringing together formal and informal knowledge. Therefore, more emphasis should be given to DUI mode in order to achieve a balance between both modes of innovation in rural areas.

There is an advisory service system around rural innovation that involves a great diversity of actors, but it is a fragmented system in which the actors are not always connected or interact with each other, which can negatively affect the process of obtaining new knowledge and the process of innovation. More research is needed to analyse and develop channels for absorbing information, codified technical knowledge and know-how in interactive innovation initiatives developed in rural environments.

Among the limitations of the study, we identified that the networks have been constructed based on reports of the case studies using an analytical framework to analyse interactive innovation initiatives, but information was not collected directly from each actor nor was a data collection tool specifically focused on analysing the interaction of the actors used, so the use of the SNA was limited. These limitations in the application of SNA could be overcome through other studies, using specific tools to directly collect the opinion of the actors involved and deepening the analysis of the interaction of the actors, taking into account elements that allow measuring the directionality, strength, intensity, or frequency of the relationship between actors. Likewise, these limitations can be addressed in future research using other approaches or tools that complement the information provided by SNA, allowing for a deeper qualitative analysis of the interactions between actors or their motives, interests, or difficulties in participating in these innovation partnerships, overcoming the limitations of information on the structure of social networks, which come from limited measures of linkages that often take a static and discrete view of something that is inherently dynamic.

Given the importance of networks in the innovation process in rural areas, it is considered necessary to develop competencies and strategies for their efficient management in order to capture the existing social capital, articulate through the actors the various

existing sectors (academia, public institutions, civil society, private firms, competitors, suppliers and customers, and others), and to cooperatively develop innovations useful for agriculture and rural development. One way can be giving greater support to interactive innovation, social learning, and user-driven innovation and not just prioritising innovation in high-tech sectors.

Finally, the impact of the conflict between Russia and Ukraine continues to affect the European Union's economy and, among other things, access to high-demand agricultural products, such as wheat and barley (both countries produce almost one third of the world's wheat and barley). In this regard, it is considered that innovation initiatives, supported by local stakeholder networks, could be an alternative to promote innovation initiatives to address the current food crisis.

**Author Contributions:** Conceptualisation, S.B.G.-O., J.M.D.-P. and J.F.N.E.; Methodology, S.B.G.-O., J.F.N.E. and J.M.D.-P.; Software, J.F.N.E. and S.B.G.-O.; Validation, J.M.D.-P. and J.F.N.E.; Formal Analysis, S.B.G.-O., J.F.N.E. and J.M.D.-P.; Writing—Original Draft Preparation, S.B.G.-O.; Writing—review and editing, J.M.D.-P. and J.F.N.E.; Visualisation, J.F.N.E. and S.B.G.-O.; Supervision, J.M.D.-P. and J.F.N.E. All authors have read and agreed to the published version of the manuscript.

**Funding:** This research received no external funding.

**Institutional Review Board Statement:** Not applicable.

**Informed Consent Statement:** Not applicable.

**Data Availability Statement:** https://liaison2020.eu/our-network/case-studies/, accessed on 15 September 2022.

**Acknowledgments:** The authors would like to thank all the researchers that worked in the seventeen case studies of H2020-funded LIAISON project of the European Commission (Grant agreement ID: 773418).

**Conflicts of Interest:** The authors declare no conflict of interest.

## Appendix A

**Table A1.** General information about 17 multi-actor partnerships for agricultural innovation.

| CS | Description | Group | Location (Countries) | Dimension | Specific Topic | Principal Funds | Leader | Types of Actors in the "Core" of Multi Actor Partnership |
|---|---|---|---|---|---|---|---|---|
| CS01 | Farmers to plant and sell hemp nuts and straw | Atlantic/North Sea | Germany | Local | Agriculture: hemp | CAP-RDP | Farmers | 4 types Farmers Market actor—supply side Support organisation Other |
| CS02 | Optimise the processing of hops and providing the best quality to local (regional) breweries | Atlantic/North Sea | Belgium | Local | Agriculture: hops farming | CAP-RDP | Farmers | 2 types Farmers R&D departments in companies |
| CS03 | Innovative beehive system | Baltic, Danube, Balkan | Bulgaria | Local | Agriculture: apiculture equipment | Private financing | Market actor—supply side | 4 types Market actor—supply side Civil society Farmers Research entity |
| CS04 | Re-introduction of traditional pasture management strategies for biodiversity conservation | Baltic, Danube, Balkan | Bulgaria | Sub-national | Agriculture: grassland management | Life | Civil society | 3 types Civil society Farmers Others |

**Table A1.** *Cont.*

| CS | Description | Group | Location (Countries) | Dimension | Specific Topic | Principal Funds | Leader | Types of Actors in the "Core" of Multi Actor Partnership |
|---|---|---|---|---|---|---|---|---|
| CS05 | Design and disseminating of home-made machinery for small-scale agriculture | Mediterranean | France | Local | Agriculture: agricultural machinery | Misscellaneous | Farmers | 1 type Farmers |
| CS06 | Production cluster and dairy farmers' research | Baltic, Danube, Balkan | Poland | Sub-national | Agriculture: wheat production and processing | CAP-RDP | Research entities | 5 types Research entities Farmers Support organisations Market actors—supply side Administrative bodies |
| CS07 | Production cluster and dairy farmers' research | Baltic, Danube, Balkan | Estonia | Sub-national | Agriculture: dairy farming | CAP-RDP | Research entities | 6 types Research entities Farmers Support organisation Market actors—supply side Market actors—demand side Others |
| CS08 | Water management and use of smart irrigation technology | Atlantic/North Sea | Switzerland | Sub-national | Agriculture: irrigation | Public financing | Research entities | 3 types Research entities Educational institutions Administrative bodies |
| CS09 | Sustainable production and marketing of wines at regional level | Mediterranean | Portugal | Sub-national | Agriculture: viticulture and processing | Misscellaneous | Support organisation | 4 types Support organisations Research entities Farmers Administrative bodies |
| CS10 | Business support, knowledge exchange, advice and training for farmers and foresters | Atlantic partners—focus IE/UK and Scandinavia | United Kingdom | Sub-national | Agriculture: support network | CAP-RDP | Farmers | 4 types Farmers Research entities Support organisations Administrative bodies |
| CS11 | Research network for organic and sustainable agriculture | Baltic, Danube, Balkan | Hungary | Sub-national | Agriculture: support network | Misscellaneous | Research entities | 6 types Research entities Farmers Educational institutions Administrative bodies Civil society Support organisations |
| CS12 | More efficient irrigation technology, specifically adopted to needs of local irrigation communities | Mediterranean | Spain | Sub-national | Agriculture: irrigation | Public funding | R&D departments in companies | 2 types Farmers R&D departments in companies |

**Table A1.** *Cont.*

| CS | Description | Group | Location (Countries) | Dimension | Specific Topic | Principal Funds | Leader | Types of Actors in the "Core" of Multi Actor Partnership |
|---|---|---|---|---|---|---|---|---|
| CS13 | Pathways to phase-out contentious inputs from organic agriculture in Europe | Atlantic/North Sea | Switzerland, Germany, Denmark, Spain, France, Greece, Italy, Norway, Poland, Sweden, Turkey, United Kingdom | Multinational | Agriculture: organic agriculture | H2020 | Research entities | 1 type Research entities |
| CS14 | Improve profitability of dairy farms changing the cow feed | Atlantic/North Sea | France, Belgium | Multinational | Agriculture: livestock feed | Interreg | Research entities | 2 types Research entities Support organisations |
| CS15 | Innovative solutions to reduce food losses in the food supply chain | Atlantic/North Sea | Netherlands, Belgium, France, United Kingdom, Ireland, Germany | Multinational | Agriculture: food waste solutions | Interreg | Farmers | 6 types Research entities Farmers Support organisations Market actors—supply side Civil society Administrative bodies |
| CS16 | Waste recycling and valorisation of agricultural waste co-products and bio products | Atlantic partners—focus IE/UK and Scandinavia | Ireland, Germany, Belgium, Italy, Spain, Greece, Croatia, United Kingdom, China. | Multinational | Agriculture: agricultural waste solutions | H2020 | Research entities | 3 types Research entities Support organisations Market actors—supply side Civil society |
| CS17 | Sustainable precision agriculture: research and knowledge | Mediterranean | Italy, Spain, Portugal, Greece | Multinational | Agriculture: learning in precision agriculture | Erasmus + | Research entities | 4 types Research entities Farmers Support organisations Market actors—supply side |

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
