# Peer review of "Multi-Actor Partnerships for Agricultural Interactive Innovation: Findings from 17 Case Studies in Europe"

_land, doi:10.3390/land11101847_

Round 1

Reviewer 1 Report

Thank you for the opportunity to read this interesting article. Authors in detail presented theoretical background, methodology, and especially results which is one of the great strengths of this work. 

This paper is one of the few where I did not find major objections and I believe that it should be published because authors deal with important topic in an appropriate way and the results can attract the attention of the readers.

Authors just should pay attention to the following:

- adjust the numbers of figures (since you have two Figures 1); 

- I would like to ask whether the authors have found anything different from the previous research. It would be more interesting if authors would indicate some differences and state the possible reasons for such differences. The inclusion of such differences would make the research much more interesting and meaningful; 

- indicate in the conclusion section limitations of your research and how they can be addressed in future research;

Author Response

We are grateful for the comments and suggestions made by you, which we consider pertinent. Your contributions have guided the process of revising and improving the paper. We hope that this new version of the paper will meet your expectations for improvement. These comments have been incorporated as described below:

General items

  • Are the conclusions supported by results? Can be improved

We added a conclusion about the limitations of the research and how they can be addressed in future research (see lines 675 - 689) and a conclusion related to the main results shown in the figures presented in the study (see lines 648 – 662). The entire section was reorganized incorporating these new conclusions.

Specific suggestions / comments

  • Adjust the number of figures

The number of figures were adjusted.

  • Indicate in the conclusion section limitations of your research and how they can be addressed in future research

We complemented the previous conclusion “Some of the challenges encountered when applying the SNA are the absence of complete network data for each case study, so it relies on egocentric data. While data on the structure of multi-actor innovation partnerships are identified, there are no data that allow for a deeper qualitative analysis of the interactions between actors, their motives, interests or difficulties in participating in these innovation partnerships. The use of other approaches or tools could complement the information provided by the SNA and overcoming the limitations of information on the structure of social networks, which come from limited measures of linkages that often take a static and discrete view of something that is inherently dynamic”, with news ideas “Among the limitations of the study, we identified that the networks have been constructed based on reports of the case studies using an analytical framework to analyse interactive innovation initiatives, but information was not collected directly from each actor nor was a data collection tool specifically focused on analysing the interaction of the actors used, so the use of the SNA was limited. These limitations in the application of SNA could be overcome through other studies, using specific tools to directly collect the opinion of the actors involved and deepening the analysis of the interaction of the actors, taking into account elements that allow measuring the directionality, strength, intensity or frequency of the relationship between actors. Likewise, these limitations can be addressed in future research using other approaches or tools that complement the information provided by SNA, allowing for a deeper qualitative analysis of the interactions between actors, their motives, interests or difficulties in participating in these innovation partnerships, overcoming the limitations of information on the structure of social networks, which come from limited measures of linkages that often take a static and discrete view of something that is inherently dynamic” (see lines 675 - 689).

  • I would like to ask whether the authors have found anything different from the previous research. It would be more interesting if authors would indicate some differences and state the possible reason for such differences.

We introduce a new reference that supports different findings of this study “It should be noted that the findings on the case studies analyzed do not reflect the current trend found on the emergence of the private agricultural input supply sector as a supplier and disseminator of new technologies [44].This could be due to the fact that these findings are not exclusively focused on the rural environment and are not developed as interactive innovations” (lines 489 -494).

Kind regards, 

Susana Guerrero

Reviewer 2 Report

Research methodology may not be sufficiently reflected. After all, the article is very well presented. It would be possible to add a reference to the drawings presented in the article in the conclusions.

Author Response

We are grateful for the comments and suggestions made by you, which we consider pertinent. Your contributions have guided the process of revising and improving the paper. We hope that this new version of the paper will meet your expectations for improvement. These comments have been incorporated as described below:

General items

  • Is the research design appropriate? Must be improved

We improve the research design description adding lines 217 – 225.

  • Area the methods adequately described? Must be improved

We improve the methodology description adding lines 217 – 225 and lines 243 – 247.

  • Are all the cited references relevant to the research? Can be improved

We added three new references (see lines 803-804,805-806,820-821).

  • Are the conclusions supported by results? Can be improved

We added a conclusion about limitations of the research and how they can be addressed in future research (see lines 675 - 689) and a conclusion related to the main results showed in the figures presented in the study (see lines 648 – 662). The entire section was reorganized incorporating these new conclusions.

Specific suggestions / comments

  • Research methodology may not be sufficiently reflected

We improve the methodology description adding lines 217 – 225 Four methodological steps were followed in this study. The first was the selection of agricultural innovation case studies previously analysed in the framework of a European project focused on interactive innovation optimization to accelerate innovation in agriculture, forestry and rural development. Subsequently, we proceeded to identify each of the participating actors, taking into account the interactions between them, actor types and roles they played in each innovation initiative. The third step was the evaluation of the identified interactions and the analysis of the data using the SNA, which was complemented with descriptive statistics. Finally, the data analysed were visualized to facilitate the process of presentation and discussion of the findings” and lines 243 – 247 “In each case study, we proceeded to identify each of the participating actors, taking into account the relationships between them, actor types and roles they played in each innovation initiative. This process was carried out taking into account two steps (i) identification of actors by interactions and (ii) identification of actor types and roles, which are described in detail as follows:”

  • Add a reference to the drawing presented in the article in the conclusions

We added a conclusion related to the main results showed in the figures presented in the study (see lines 648 – 662)

“We identified a diversity of actors and their roles in the innovation process, according to four interactions analysed. In relation to the funding mechanisms, the central actors are the administrative bodies with the role of funder mainly, followed by the research entities with their roles of technical expert and coordinator who contributed significantly to the formulation of the innovation initiative, and the farmers as co-funder of the innovation initiatives. At the core of the multi-stakeholder partnership, the central actors who participated in the "co-creation" of innovations are farmers and research entities with the main role of case study field workers and technical expert, respectively. Regarding to the relation with the external actors of the multi-stakeholder partnership, the central actors are research entities, followed by support organizations, administrative bodies and others, who have contributed in specific ways to the innovation, from their expertise, helping in its design, using it to validate it or disseminating it. Finally, related to the relation with the context, the central actors are the support organizations and others and their relationship with the innovation is referred to the use they make of the innovations or their diffusion”.

Kind regards, 

Susana Guerrero

Reviewer 3 Report

it's not necessary

Author Response

We are grateful for the comments and suggestions made by you, which we consider pertinent. Your contributions have guided the process of revising and improving the paper. We hope that this new version of the paper will meet your expectations for improvement. These comments have been incorporated as described below:

General items

  • Are all the cited references relevant to the research? Can be improved

We added three new references (see lines 803-804,805-806,820-821).